# Performance of five dynamic models in predicting tuberculosis incidence in three prisons in Thailand

**Nithinan Mahawan**[1], **Thanapoom Rattananupong**[2], **Puchong Sri-Uam**[3], **Wiroj Jiamjarasrangsi**[2]*

**1** School of Nursing, the Excellence Center of Community Health Promotion, Walailak University, Nakhon Si Thammarat, Thailand, **2** Department of Preventive and Social Medicine, Faculty of Medicine, Chulalongkorn University, Bangkok, Thailand, **3** Center for Safety, Health and Environment of Chulalongkorn University, Bangkok, Thailand

\* wjiamja@gmail.com

**Data Availability Statement:** All protocol and dataset files are available from the protocols.io database (https://www.protocols.io/view/

## Abstract

This study examined the ability of the following five dynamic models for predicting pulmonary tuberculosis (PTB) incidence in a prison setting: the Wells–Riley equation, two Rudnick & Milton-proposed models based on air changes per hour and liters per second per person, the Issarow et al. model, and the applied susceptible-exposed-infected-recovered (SEIR) tuberculosis (TB) transmission model. This 1-year prospective cohort study employed 985 cells from three Thai prisons (one prison with 652 cells as the in-sample, and two prisons with 333 cells as the out-of-sample). The baseline risk of TB transmission for each cell was assessed using the five dynamic models, and the future PTB incidence was calculated as the number of new PTB cases per cell and the number of new PTB cases per 1,000 person-years (incidence rate). The performance of the dynamic models was assessed by a four-step standard assessment procedure (including model specification tests, in-sample model fitting, internal validation, and external validation) based on the Negative Binomial Regression model. A 1% increase in baseline TB transmission probability was associated with a 3%–7% increase in future PTB incidence rate, depending on the dynamic model. The Wells–Riley model exhibited the best performance in terms of both internal and external validity. Poor goodness-of-fit was observed in all dynamic models (chi-squared goodness-of-fit tests of 70.75–305.1, 8 degrees of freedom, p < .001). In conclusion, the Wells–Riley model was the most appropriate dynamic model, especially for large-scale investigations, due to its fewer parameter requirements. Further research is needed to confirm our findings and gather more data to improve these dynamic models.

## Introduction

The global prevalence and incidence of *Mycobacterium tuberculosis* infection are markedly higher among prison inmates than in the general population [1]. Hence, a few recent studies have begun to focus on tuberculosis (TB) incidence in prisons [2–4]. Furthermore, the number

performance-of-five-dynamic-models-in-predicting-t-dc632zgn.html) (DOI: 10.17504/protocols.io.n92ld8z8xv5b/v1) and the Figshare repository (DOI: 10.6084/m9.figshare.26787874).

**Funding:** - WJ and NM received the 90th Anniversary of Chulalongkorn University Fund (grant no. 035, 1/2021) and the FY2021 Thesis Grant for Doctoral Degree Study of the NationalmResearch Council of Thailand (NRCT; grant no. N41D640002, 2021). - URI (the 90th Anniversary of Chulalongkorn University Fund): https://www.grad.chula.ac.th/index.php?lang=th-URI: https://www.nrct.go.th/ The funders had no role in study design, data collection and analysis, decision to publish, or preparation of the manuscript.

**Competing interests:** The authors have declared that no competing interest exist.

of incarcerated people has continuously increased globally [5]. Since prisons are hot-spots or reservoirs for TB, which can spread to the surrounding communities [6], the high incidence of TB in prisons may partly explain why global TB incidence has declined slowly in recent years (at a rate of 1%–2% per year) [7] despite enhanced control efforts, substantial research advances, and increased funding [8]. Therefore, TB control in prisons is essential for making TB-free world a reality [9]. However, due to security and feasibility reasons, most TB control measures in prisons have focused on early diagnosis [6], with less emphasis on the fundamental determinants of transmission [2, 10].

Dynamic models of pulmonary tuberculosis (PTB) transmission have been employed to evaluate the impact of various intervention measures (including structural changes) on reducing TB incidence in prisons. These include the classic Wells–Riley equation [11, 12], Rudnick & Milton's and Issarow et al.'s modified Wells–Riley equations [13, 14], and the applied susceptible-exposed-infected-recovered (SEIR) tuberculosis transmission model [15]. All these models comprise two components: the estimation of the intake dose of the infectious agent and the estimation of probability of infection at a specific intake dose [16]. The classic Wells–Riley equation was the original equation. The remainders of the three equations are modified models that have been increasingly modified to improve the accuracy of predicting the TB transmission probability. Wells–Riley's, Rudnick & Milton's, and Issarow et al.'s equations and/or models share similar concepts in their approach to the airborne transmission of infectious disease as a physical transport problem, with some minimal differences [6]. Conversely, the SEIR model also integrates the epidemiological concept of human TB transmission dynamics into the Wells–Riley model [11].

All these models have been employed to predict the potential efficacy of various intervention alternatives for managing PTB in prisoners, and the most feasible alternative was subsequently implemented. However, the improved prediction accuracy of the latter compared to the former models has never been assessed in a real-world setting. Therefore, we aimed to compare the relative effectiveness of these five dynamic models (namely, the classic Wells–Riley equation, Rudnick & Milton's and Issarow et al.'s modified Wells–Riley equations, and the applied SEIR model) in predicting the incidence rate (IR) of PTB among the inmates of three prisons in Thailand.

## Materials and methods

### The dynamic models used in estimating TB transmission probability in prisons

This section provides a concise overview of the dynamic models that were employed in estimating TB transmission probability in a prison setting and will be used in this study. It commences with the simplest model and progresses to more complex models.

**The classic Wells–Riley model.** The initial equation used to quantify the risk of airborne infection for TB and measles was the classic Wells–Riley equation, developed by Wells and Riley [17]; the equation is as follows:

$$Pi = \frac{C}{S} = 1 - exp(-Ipqt/Q)$$

It considers the intake dose of airborne pathogens in terms of the number of quanta (i.e., infectious dose) and assumes the number of randomly distributed discrete infectious particles in the air. This equation estimates the probability of infection ($Pi$) or the number of infectious cases ($C$) based on the data regarding the number of susceptible persons ($S$), the number of infectors ($I$), the pulmonary ventilation rate of a person ($p$) (0.36 m$^3$/h), the quanta generation

rate ($q$) (1 quanta/h), the exposure time interval ($t$) (month), and the room ventilation rate with clean air ($Q$) (air changes/h). Its simplicity is its advantage; however, its limitations include assumptions about (1) a well-mixed airspace and (2) a steady-state condition, which may not always be the case [16].

**Rudnick and Milton's modified model.**   Rudnick and Milton expanded the practical application of the Well–Riley model by not presuming steady-state conditions [10, 13]. As human respiration is the primary source of $CO_2$ within most indoor spaces, indoor $CO_2$ concentration could serve as a biomarker for exhaled breath that has the potential to carry infectious aerosols. Based on this concept, Milton and Rudnick devised a nonsteady-state version of the Wells–Riley equation. They used the exhaled air volume fraction to estimate the number of quanta to which susceptible people are exposed [13]. However, the assumption of a well-mixed airspace remains necessary, and the term "quanta" continues to be used as a representation of infectious dose. The revised Well–Riley equation is as follows:

$$Pi = 1 - \exp\left[-\frac{Iqpt}{Q\theta}\left\{1 - \frac{V}{Q\theta}\left[1 - exp\left(-\frac{Q\theta}{V}\right)\right]\right\}\right]$$

According to the equation, the probability of infection ($Pi$) is a function of the number of infectious individuals in a room ($I$), respiration rate (p) (360 L/h or 0.36 m$^3$/h), quantum generation rate by an infected person (q) (1.25 quanta/h for smear-positive and 0.20 quanta/h for smear-negative), total exposure time (t) (day), volume of the confined space (V) (m$^3$), time elapsed from when the room becomes occupied ($\theta$) (h), and germ-free ventilation rate (Q) (air changes/h or L/s/person) [10, 13]. Germ-free ventilation can be quantified in terms of area ventilation as air change per hour (ACH) or liters per second per person (L/s/person) [10].

**Issarow, Mulder, and Wood' modified model.**   By monitoring the air exhaled by infectors in a confined space, Issarow et al. further modified the Wells–Riley equation to flexibly estimate the risk of PTB under both steady-state and nonsteady-state conditions [14, 18]. In the estimation of transmission probability, they did not presume a well-mixed air space; rather, they considered the duration of exposure and proximity to the infector. Furthermore, because not all infectious particles will reach the target infection site, they replaced the term "quanta" with multiple parameters (including surviving airborne infectious doses and alveoli deposition fraction) that considered that the infectious particles that initiate the infection are determined by the respiratory deposition fraction, which is the probability of each infectious particle reaching the target infection site of the respiratory tract and causing infection. The equation is as follows:

$$Pi = 1 - \exp(-Pv(\beta - \mu)\theta pft),$$

where Pi denotes the probability of infection; Pv is the prevalence of infectors in the space; $\beta$ represents the total airborne infectious particles generation rate released by an infector(particles/s); $\mu$ is the mortality rate of generated airborne infectious particles by the infector that do not reach the alveolar (particles/s); ($\beta - \mu$) represents the doses of surviving airborne infectious strain per unit time (doses/h) that reach the alveoli to establish infection depending upon the host immune systems and the virulence of the infecting strain [14]; $\theta$ is the deposition fraction of infectious doses in the alveoli; p is the respiration rate (0.36 m$^3$/h); t is the duration of exposure (h); and $f$ or pn/Q is the fraction of rebreathed air. In pn/Q, p signifies the average pulmonary ventilation rate (0.36 m$^3$/h); n is the susceptible inmate; and Q denotes the ventilation rate (ACH) [18].

**The applied SEIR model.**   Noakes et al. developed the applied SEIR model after evaluating the limitations of the standard Wells–Riley model, which only describes the number of

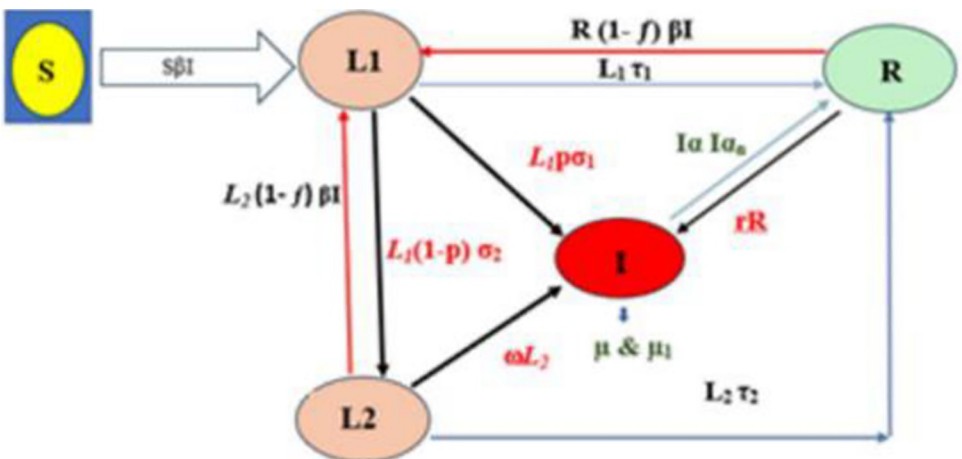

**Fig 1. Applied SEIR model [15].**

infections for a set number of infectors rather than the whole dynamics of an epidemic [19]. They therefore integrated the Wells–Riley model with the classic SEIR model, which includes an incubation period and consists of systems of first-order differential equations that delineate the progression from susceptible (S) to infectious (I) and to recovered or removed (R) individuals (Fig 1). This combination enables the quantification of the effective contact or transmission rate (β) between infectors and those susceptible in relation to room ventilation, environmental conditions, and airborne infectious material levels in the space.

Fig 1 depicts the division of the inmate population in a prison into five categories based on their TB status: susceptible (S), short-term latently infected (L1), long-term latently infected (L2), infectious (I), and recovered (R). S is susceptible (person); L1 = short-term latently infected (persons/year); L2 = long-term latently infected (persons/year); I = infectious (percent/180 days); R = recovered (persons/year); β = transmission rate (per person per year); Ω = inmates' turnover (percent/year); μI = TB-related death (per year); p = proportion of fast progressors (percent); σ1 = rate fast progressors develop infectious TB (primary progression) (per year); σ2 = rate fast progressors move to slow progressors (per year); ω = rate slow progressors develop infectious TB (reactivation) (per year); r = relapse rate (per year); f = partially acquired immunity following primary infection for treated persons (percent); τ1, τ2 = rate of recovery under preventive therapy for fast/slow progressors (per year); α = rate of recovery under anti-tuberculosis treatment (percent); and αn = natural recovery (per year).

## Study population

A 1-year prospective cohort study was conducted to observe the occurrence and number of new PTB cases among the study samples from 985 cells in three prisons in Thailand, with cells in prisons serving as the unit of analysis. The three prisons had a total of 17,557 inmates. Two prisons were in Bangkok, and one prison was in Rayong Province (180 km east of Bangkok). One of the Bangkok prisons was a super-maximum-security prison, with 239 cells in five zones and 4,734 inmates (prison A), whereas the other was a maximum-security prison with 652 cells in six zones and 6,607 inmates (prison B). Rayong Province's prison was a maximum-security facility with 94 cells in four zones and 6,216 inmates (prison C). PTB is considered a recurrent disease; therefore, all 17,557 inmates included in the study were still at risk, even if they had previously contracted the disease. The study was conducted with the approval of the Ethical Review Board of Chulalongkorn University Faculty of Medicine (IRB No. 610/63) and

permission from the Department of Corrections at the Ministry of Justice. Before the study commenced, the superintendent of each prison read the "Information Sheet for Participants" and signed the "Submission Agreement for Volunteers." Consent from individual detainees was not obtained, as the investigation did not gather information that could be used to identify inmates.

## Exposure assessment

Baseline risk assessment of TB transmission was conducted for the inmates of each cell using each of the five dynamic models: the classic Wells–Riley equation [17], Rudnick & Milton's, and Issarow et al.'s modified Wells–Riley equations [13, 14], and the applied SEIR model [20]. The detailed procedure has been described previously [21]. In brief, the information regarding the architectural and environmental characteristics of the cells and the absolute ventilation rate (Q; L/s) was collected by trained inmates from the corresponding zone or prison and the ventilation rate in ACH was collected by the principal investigator (NM) and a coinvestigator (PS), while the data about the demographic composition and health status of its inmates were obtained from primary and secondary data sources from each zone in the prisons. The data on PTB infection was obtained from the computer and document databases of the corresponding zonal and central administrative and medical facilities in the prisons. The data on TB progression-related parameters and physiological parameters was obtained from relevant international literature. The parameters of the efficacy of PTB treatment were obtained by utilizing the databases of the study prisons. The risk of TB transmission for each cell was subsequently estimated based on the five previously mentioned dynamic models.

## Outcome assessment

For practical considerations, the follow-up periods in each prison were not initiated simultaneously. For prisons A and B, the follow-up period was from January 2021 to February 2022, and for prison C the follow-up period was from April 2021 to April 2022. Because the standard practice in all prisons is to always maintain full occupancy in all cells, the number of person-years at risk allocated for each cell was derived by multiplying the number of residing inmates by the number of follow-up years.

The outcome of interest was PTB occurrence, as defined by the 9th and 10th Revisions of the International Classification of Diseases (ICD 9 and 10) through codes A15.0 (tuberculosis of the lung confirmed by sputum microscopy with or without culture), A15.1 (tuberculosis of the lung confirmed by culture only), and A16.0 (tuberculosis of the lung, bacteriologically and histologically negative). This did not include extrapulmonary TB. PTB cases were diagnosed by the physicians of the hospitals under the Department of Corrections at the Ministry of Justice (prisons A and B) and the Ministry of Public Health (prison C). The diagnostic procedure adhered to accepted clinical practice and included a physical examination, chest radiograph, sputum acid-fast bacilli smear, and nucleic acid amplification testing using the Xpert® MTB/RIF assay (Cepheid, Sunnyvale, USA) [22]. The incidence of PTB outcome was regarded as a repeated outcome, meaning that all new PTB case(s) in each cell that occurred during the follow-up period for the individual(s) of each cell were considered, even among those with a history of previously cured PTB. The data sources used for determining this outcome included documents and computer databases of the central administrative and medical facilities in the prisons, while information regarding the cell location of the PTB inmates was determined by surveys and interviews with zonal staff and inmates who were prison health volunteers. The data included the number of TB-infected patients in each cell and zone, which represented the incidence at the cell and zonal levels. This data was recorded from January 2021 to February

2022 for prisons A and B, and from April 2021 to April 2022 for prison C. The magnitude of PTB was reflected by the number of PTB cases per cell and the IR of PTB, which was calculated as the number of new PTB cases divided by the person-years of observation and reported as the number of new PTB cases per 1,000 person-years.

## Statistical analysis

A four-step procedure, which encompassed model specification tests, in-sample model fitting, internal validation, and external validation, was implemented to evaluate the performance of five dynamic models. This procedure was implemented in accordance with the method outlined by Dep et al. [23] and was adapted from the method used in research conducted by Holodinsky et al. and Le et al. [24, 25]. For this process, the overall dataset was divided into two-thirds to be used as a training dataset (prison B data with n = 652, referred to as the in-sample) and one-third as a test dataset (prison A and C data with n = 333, referred to as the out-of-sample); the standard rule of thumb [26]. The initial three steps were performed employing the in-sample as a larger sample can generate more stable prediction equation with less variance, and the final step was performed utilizing the out-of-sample. The outcome variable was the number of PTB cases, while the predictors were the TB transmission probabilities estimated by the five dynamic models. The subsequent sections contain specifics regarding each procedure.

**Model specification tests.**   The PTB outcome is a nonnegative integer value ranging from 0 to 5 cases with substantial mass at zero (85.99%) and a highly skewed distribution with over-dispersion (mean of 0.2 cases and variance of 0.35). To determine the optimal analytical methodology, the association between baseline TB transmission probability and future PTB incidence was investigated using a combination of diverse count models and their respective Akaike information criterion (AIC) and Bayesian information criterion (BIC) statistics (S1 Table). Inferring the reduced AIC and BIC values as suggestions for a more optimal model [27], the results demonstrated that the preferred count models are the Negative Binomial Regression Model (NBRM), Zero-inflated Poisson Regression Model (ZIP), and Zero-Inflated Negative Binomial regression model subject to the type of TB transmission probability predictor in each analysis. The NBRM was deemed the most appropriate count model for this analysis because of its relatively straightforward nature that facilitates the analyses in later steps.

The Ramsey's Regression Specification Error Test was subsequently implemented to examine the necessity for higher-order predictors. The results indicated that the NBRM was accurately specified without the inclusion of higher-order predictors.

**In-sample model fitting.**   To ascertain the association between the TB transmission probability at baseline and the future occurrence of PTB cases, NBRM was conducted for each dynamic model separately, with the corresponding TB transmission probability serving as the sole predictor. The incidence rate ratio (IRR) and 95% confidence interval (CI) were employed as measures of association. Additionally, graphical representations were provided to illustrate the correlation between the probability of TB transmission and the future IR of PTB.

The performance of the dynamic models was then evaluated using three approaches. The first approach compared the mean predicted probabilities and observed proportions of each count of PTB cases, ranging from 0 to 9. A Chi-squared goodness-of-fit test was then conducted, and the highly significant test statistic indicated a model with poor fit [26]. The second approach computed additional measures of goodness-of-fit, including root mean square error (RMSE), mean absolute error (MAE), and bias [24, 25]. These three measures capture the bias between predicted probabilities and observed proportions for each count of the dynamic models. The smaller the bias, the better the model. The third approach used the AIC and the BIC for model discrimination among non-nested models. Low values in these tests indicated more

parsimonious models [27]. A difference in AIC value of two or more was considered significant [27]. For BIC, a difference in BIC value between two models of more than 10 indicated very strong evidence; a value of 6–10 indicated strong evidence; a value of 2–6 indicated positive evidence; and a value of 0–2 indicated weak evidence for the difference in model performance [28, 29].

**Internal validation.**   In-sample model checks were conducted through 10-fold repeated cross-validation. This process randomly portioned the in-sample into ten subsamples, with a single subsample retained as the validation data for testing the model and the remaining nine subsamples used as training data. This cross-validation procedure is subsequently repeated ten times (the folds), with each of the ten subsamples being used as the validation data exactly once. Each time, log likelihood was employed to compare models, with a higher value indicating more desirable models [23].

**External validation.**   The predicted or expected PTB cases for each cell in the out-of-sample dataset were obtained by applying the resultant fitted models from the in-sample dataset to the out-of-sample dataset. Model accuracy was subsequently determined by calculating the RMSE, MAE, and bias. Calibration plots were also implemented to evaluate the model's calibration. A precisely calibrated model's calibration curve would be in close alignment with the 45-degree diagonal line on the plot. The calibration slope depicts a target value of 1. Calibration slope <1 indicates that the estimated risks are excessively extreme (i.e., too high for cell at high risk and too low for cell at low risk), while a slope > 1 suggests the opposite, i.e., the risk estimates are too moderate [30]. The target value of the calibration intercept is 0. Negative values indicate overestimation, while positive values indicate underestimation.

Finally, sensitivity analysis was performed to evaluate the robustness regarding the findings of the relative performance of the TB transmission probability models. This was achieved by alternating the dataset between the in-sample and out-of-sample. Subsequently, the procedures described previously for model fitting and internal and external validations were repeated.

The statistical significance level was set at 0.05 for all analyses. The STATA version 17.0 software (Stata SE–Standard Edition 17.0 Stata Statistical Software: Release 17.0 College Station, TX: StataCorp LLC) was used to perform all data analyses.

## Results

### Characteristics of the cells in the study

During the follow-up period, at least one new PTB case occurred in 138 of the 985 cells (14.01%), with the number of new PTB cases ranging from 1 to 5 cases in each cell. This amounted to 201 new PTB cases among a total of 17,557 inmates. The overall PTB IR (95% CI) was 10.78 (9.33–12.37) per 1,000 person-years.

The baseline characteristics of the cells in the in-sample were significantly different from those in the out-of-sample in terms of architectural characteristics and the demographic characteristics and health composition of the cell inmates (Table 1). These had led to substantially higher estimated TB transmission probabilities for the latter group than for the former. The out-of-sample group exhibited a higher mean and variance of new PTB cases per cell (0.36 and 0.67) than the in-sample group (0.12 and 0.18) during the follow-up period, which was indicative of its greater overdispersion than the latter group. However, its overall PTB IR was, lower than that of the in-sample group (Table 1).

### In-sample model fitting

Fig 2 and Table 2 illustrate the association between the estimated TB transmission probability (independent variable) and the observed PTB IR (dependent variable) using NBRM. A higher

**Table 1. Comparison of the cell characteristics between the in-sample and out-of-sample.**

| Characteristics | In-sample (Prison B, n = 652) | | Out-of-sample (Prisons A&C, n = 333) | | p |
|---|---|---|---|---|---|
| | Med | (Q1, Q3) | Med | (Q1, Q3) | |
| **AT BASELINE** | | | | | |
| **Architectural characteristics** | | | | | |
| Floor area (m$^2$) [†] | 7.71 | (7.71, 16.44) | 28.00 | (28.00, 40.49) | a |
| Ceiling height (m) [†] | 3.48 | (3.35, 3.50) | 3.60 | (2.80, 3.60) | a |
| Cell volume (m$^3$) [†] | 26.98 | (25.82, 57.22) | 100.80 | (100.8, 111.35) | a |
| Ventilation rate (ACH) [†] | 28.93 | (17.24, 46.86) | 31.08 | (22.56, 44.92) | a |
| Absolute ventilation rate (L/s/p) [†] | 24.53 | (19.19, 30.86) | 25.87 | (20,00, 40.94) | a |
| **Demographic characteristics and health composition of the cell inmates** | | | | | |
| Number of inmates in cell [†] | 6 | (5, 11) | 24.00 | (22, 41) | a |
| Area per person (m$^3$/person) [†] | 5.20 | (4.77, 5.63) | 4.20 | (3.13, 4.58) | a |
| Inmate turnover rate (%/year) [†] | 1.69 | (1.39, 6.86) | 9.27 | (6.18, 21.9) | a |
| Time to diagnosis of TB case in cell (days) [†] | 116.39 | (70.85, 116.39) | 298.57 | (161.93, 446.53) | a |
| Number of overall TB cases in cells (prevalence in 180 days) ℗n (%)] [‡] | | | | | a |
| 0 | 597 | (91.56) | 259.00 | 77.78 | |
| 1+ | 55 | (8.44) | 74.00 | 22.23 | |
| The prevalence of infectious patient in zone (prevalence per 100 persons per180 days) [†] | 1.32 | (0.49, 1.32) | 0.98 | (0.53, 1.72) | a |
| **TB transmission probability** [†] | | | | | |
| The Wells–Riley model | 0.036 | (0.018, 0.115) | 0.100 | (0.060, 0.282) | a |
| Rudnick and Milton-proposed models (ACH) | 0.040 | (0.020, 0.085) | 0.135 | (0.046, 0.257) | a |
| Rudnick and Milton-proposed models (L/s/p) | 0.019 | (0.006, 0.038) | 0.044 | (0.017, 0.099) | a |
| Issarow et al.-proposed models | 0.005 | (0.002, 0.037) | 0.051 | (0 .014, 0.129) | a |
| SEIR TB transmission model | 0.181 | (0.178, 0.250 | 0.243 | (0.219, 0.253) | a |
| **DURING FOLLOW-UP** | | | | | |
| **PTB incidence** | | | | | |
| Number per cell; Median (Q1, Q3) | 0 | (0, 0) | 0 | (0, 0) | a |
| Number per cell; Mean (SD) | 0.12 | (0.42) | 0.36 | (0.82) | |
| Incidence rate per 1,000 person-years | | | | | |
| Incidence rate (95% CI) | 11.49 | (9.26, 14.27) | 10.33 | (8.63, 12.37) | a |

ACH = Air change per hour. L/s/p = Liter per second per person. CI = Confidence interval. Med (Q1, Q3) = Median (Quartile 1, Quartile 3). SD = Standard deviation. PTB = Pulmonary tuberculosis. TB = Tuberculosis.

[a] statistical significance level of <0.05 between the in-sample and out-of-sample.

[†]Two-sample Wilcoxon rank-sum (Mann–Whitney) test was used to compare the difference between the medians.

[‡]Chi-square (or Fisher's exact) test.

The prevalence of infectious patients in the zone was determined by dividing the number of TB cases in the zone by the number of inmates in the zone. This data was recorded as a percentage per six months from July to December 2020 for prisons A and B and from October 2020 to March 2021 for prison C.

TB transmission probability at baseline was associated with a progressive increase in the future PTB IR for all dynamic models (Fig 2). A 1% increase in the estimated TB transmission probability was correlated with a 3%–7% increase in the PTB IR. The magnitude of increase was the highest for the two Rudnick and Milton's models and comparable among the remaining models (Table 2).

The observed data span differed among the dynamic models: widest for the Issarow et al. model, followed by the classic Wells–Riley equation, the Rudnick & Milton's (ACH) model,

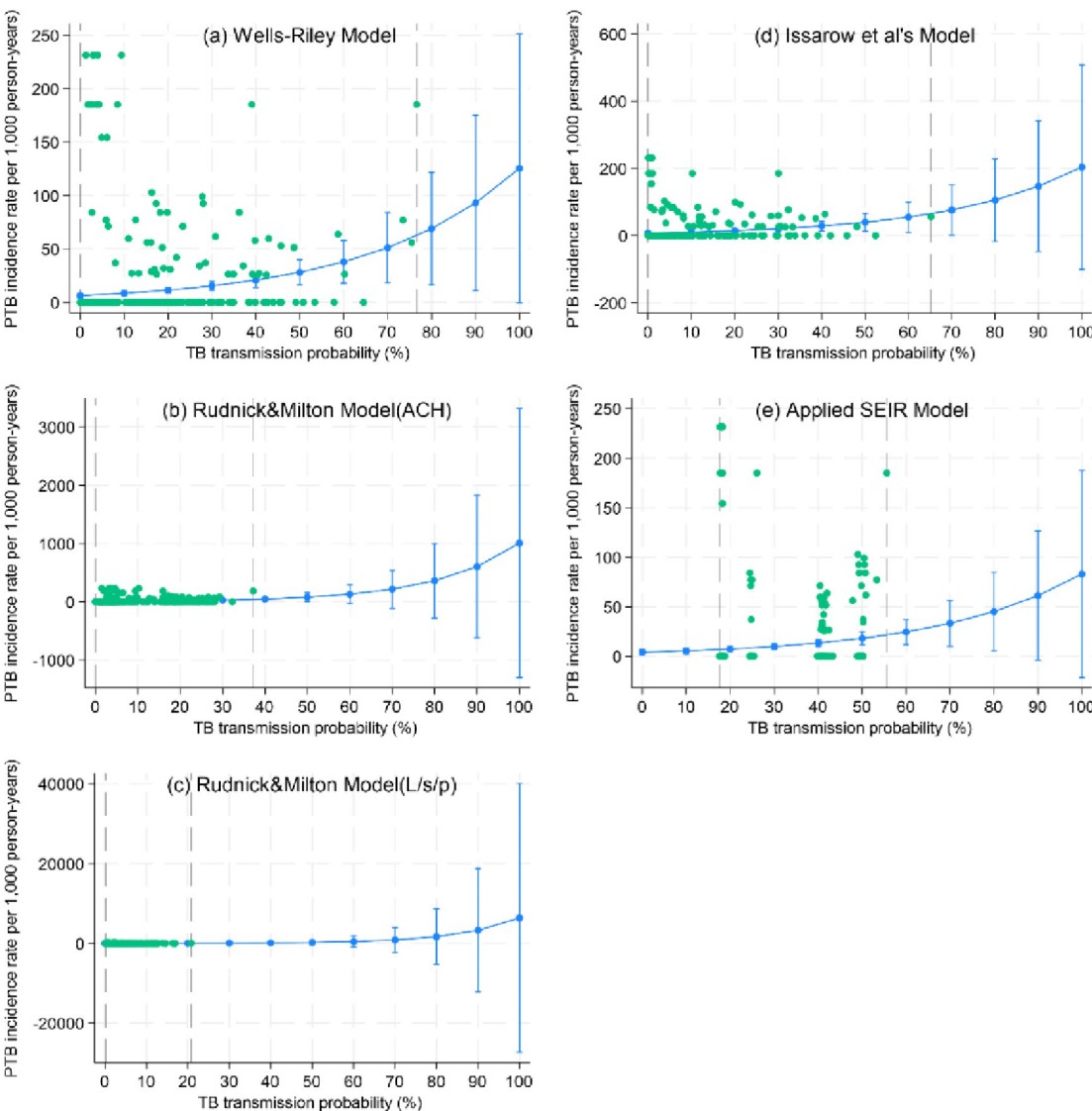

**Fig 2. Distribution of the PTB IR based on the TB transmission probability estimated using the five dynamic models.** The green scatter plots indicate the observed data for individual prison cells. The dark blue line indicates the fitted line. The two vertical dashed lines represent the fitted data interval.

the applied SEIR model, and narrowest for the Rudnick & Milton's (L/s/p) model (Fig 2 and Table 2). The PTB IR predictions within the observed data range were comparable and higher for the Wells–Riley and Issarow et al. models and comparably lower for the remainder of the models (S1 Fig). However, the predictions beyond this range exhibited significant uncertainty, with the highest values for the two Rudnick & Milton's models, followed by the Issarow et al. model, the Wells–Riley equation, and the applied SEIR model, respectively (Fig 2).

## In-sample goodness-of-fit tests

The Wells–Riley model outperformed the other models in predicting the in-sample PTB IR, as evidenced by the lowest values of the goodness-of-fit and discrimination test parameters, as shown in Table 3 and S2 Table. This was followed by the Issarow et al.'s and the applied SEIR

**Table 2. Fitted models of the negative binomial regression (NBRG) for the five dynamic models (n = 652).**

| Prediction Model | Range of observed probability (%) (x1) | Constant (b0) | Beta (b1) | SD | IRR | (95%CI) |
|---|---|---|---|---|---|---|
| Wells–Riley | 0 to 77.6 | −5.067 | 0.030 | 0.006 | 1.030 | (1.018, 1.043) |
| Rudnick&Milton(ACH) | 0 to 53.3 | −5.116 | 0.051 | 0.013 | 1.053 | (1.025, 1.081) |
| Rudnick&Milton(L/s/p) | 0.1 to 35.5 | −4.823 | 0.067 | 0.028 | 1.069 | (1.011, 1.130) |
| Issarow et al. | 0 to 88.4 | −4.875 | 0.033 | 0.009 | 1.033 | (1.016, 1.051) |
| Applied SEIR | 15.6 to 57.0 | −5.544 | 0.031 | 0.010 | 1.031 | (1.011, 1.051) |

CI = Confidence interval, IRR = Incidence rate ratio, SD = standard deviation

Equation used for prediction: $C_j = \exp[\ln(E_j) + b_0 + b_1 x_{1j}]$, where C = expected number of PTB cases, j = $j^{th}$ cell of the prison, E = the exposure of person-years of observation, b0 = constant or intercept, b1 = beta coefficient or slope, x1 = TB transmission probability estimated by the specified dynamic model, exp = exponential function, ln = natural logarithm.

models, while the Rudnick & Milton-L/s/p model performed the worst. However, the chi-squared goodness-of-fit tests were statistically significant for all dynamic models, indicating that these models were poor fits. The IR of PTB was underpredicted by the Rudnick & Milton-L/s/p and applied SEIR models (negative bias values), while the other models performed as expected.

## Internal validation

To facilitate interpretation, the Well–Riley model is employed as the base model in the 10-fold cross-validation, as the results of the previous phase indicated that it was the best fit. The overall result was consistent with the previous step, i.e., the Wells–Riley model was the best fit, followed by the Issarow et al. model, while the worst fit model was the Rudnick & Milton (L/s/p) model (Fig 3).

## External validation

The Wells–Riley model is the best fit model, which is in accordance with the preceding procedures. However, its RMSE, MAE, and bias values were the lowest; the calibration slope was closer to 1 and the intercept closer to zero [Table 4 and Fig 4A]. In contrast, Issarow et al.'s model exceedingly overpredicted the PTB incidence, as evidenced by its slope being much lower than 1 [Fig 4D]. The incidence of PTB was also overpredicted by the two Rudnick & Milton models and the applied SEIR models, albeit to a significantly lesser extent [Table 4 and Fig 4B, 4C and 4E].

CI represents the confidence interval. The blue line indicates the fit line, and the gray shading represents its 95% confidence interval. The dot line represents the ideal fitted line. Black hollow circles represent the observed values.

**Table 3. In-sample goodness-of-fit and discrimination test results of five dynamic models.**

| Prediction model | Goodness-of-fit test results | | | Discrimination tests | |
|---|---|---|---|---|---|
| | RMSE | MAE | Bias | AIC | BIC |
| Wells–Riley | 0.379 | 0.182 | 0.002 | 424.201 | 437.641 |
| Rudnick & Milton(ACH) | 0.382 | 0.188 | 0.000 | 430.719 | 444.159 |
| Rudnick & Milton(L/s/p) | 0.385 | 0.192 | −0.001 | 439.714 | 453.154 |
| Issarow et al. | 0.384 | 0.185 | 0.003 | 430.402 | 443.842 |
| Applied SEIR | 0.384 | 0.185 | −0.002 | 430.402 | 443.842 |

RMSE = Root Mean Square Error, MAE = Mean absolute error, AIC = Akaike information criterion, BIC = Bayesian information criterion

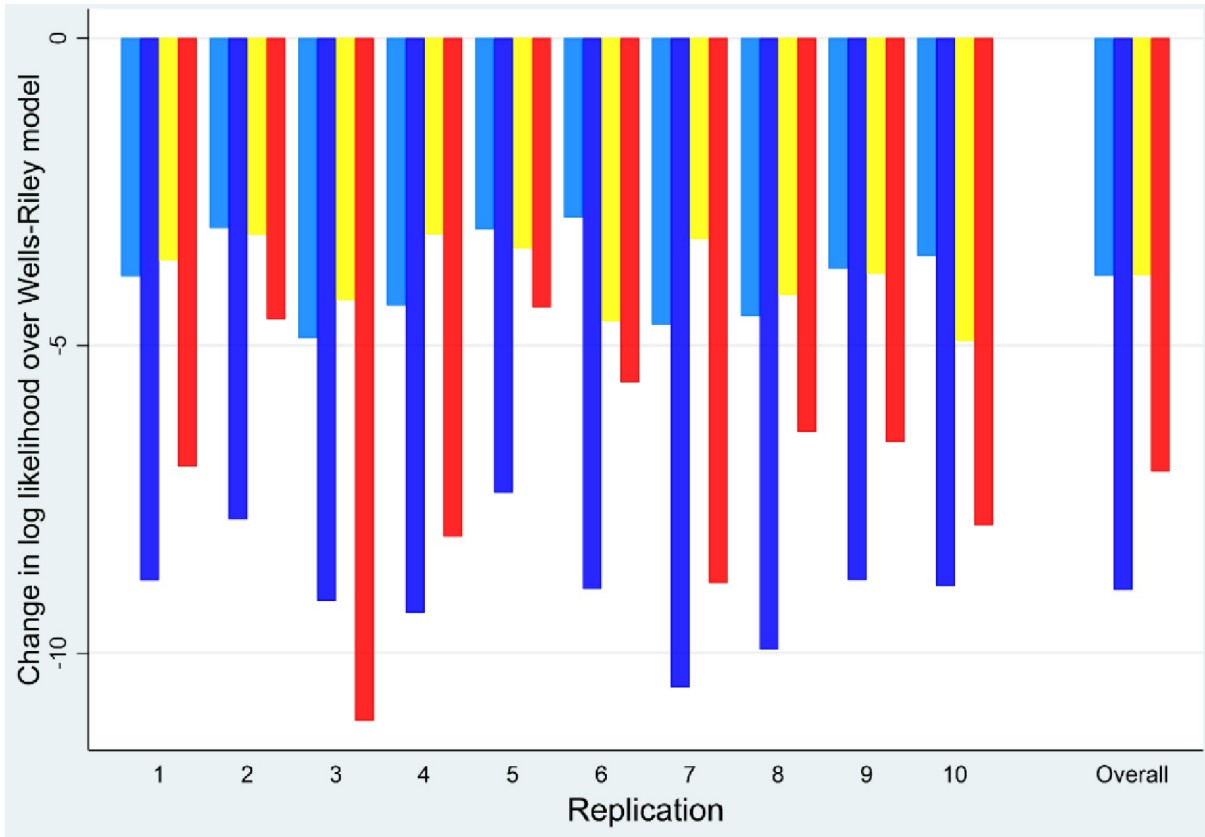

**Fig 3. The 10-fold cross-validation results of four dynamic models compared to the Wells–Riley model.** Light blue represents Rudnick & Milton (ACH), blue—Rudnick & Milton (L/s/p) model, yellow—Issarow et al.'s model, and red—Applied SEIR model.

## Sensitivity analysis

The reanalysis of the relationship between baseline TB transmission probability and future PTB incidence after switching the dataset between the in-sample and out-of-sample revealed that the beta coefficients and the corresponding IRRs representing were slightly lower than those in the main analysis for all dynamic models. Additionally, the statistically significant and relative magnitudes of these two parameters among the dynamic models were largely maintained (S3 Table). The sensitivity analysis also confirmed the Well–Riley model's superior

**Table 4. Comparison of accuracy and bias metrics from external validation (out-of-sample, n = 333).**

| Prediction model | Range of observed probability (%) (x1′) | RMSE | MAE | Bias | Min predicted value | Max predicted value | Calibration | |
|---|---|---|---|---|---|---|---|---|
| | | | | | | | slope | intercept |
| Wells–Riley | 0.4–71.8 | 0.701 | 0.461 | −0.010 | 0.023 | 4.218 | 1.102 | −0.025 |
| Rudnick & Milton(ACH) | 0.6–53.3 | 0.732 | 0.511 | 0.112 | 0.029 | 3.528 | 0.848 | −0.041 |
| Rudnick & Milton(L/s/p) | 0.1–35.5 | 0.756 | 0.534 | 0.120 | 0.010 | 5.463 | 0.672 | 0.037 |
| Issarow et al. | 0.1–88.4 | 3.776 | 0.728 | 0.326 | 0.008 | 70.541 | 0.085 | 0.299 |
| Applied SEIR | 15.6–57.0 | 0.805 | 0.513 | 0.104 | 0.023 | 9.473 | 0.519 | 0.118 |

RMSE = Root Mean Square Error, MAE = Mean absolute error

x1′ = TB transmission probability at baseline of the out-of-sample, which was estimated by the specified dynamic model

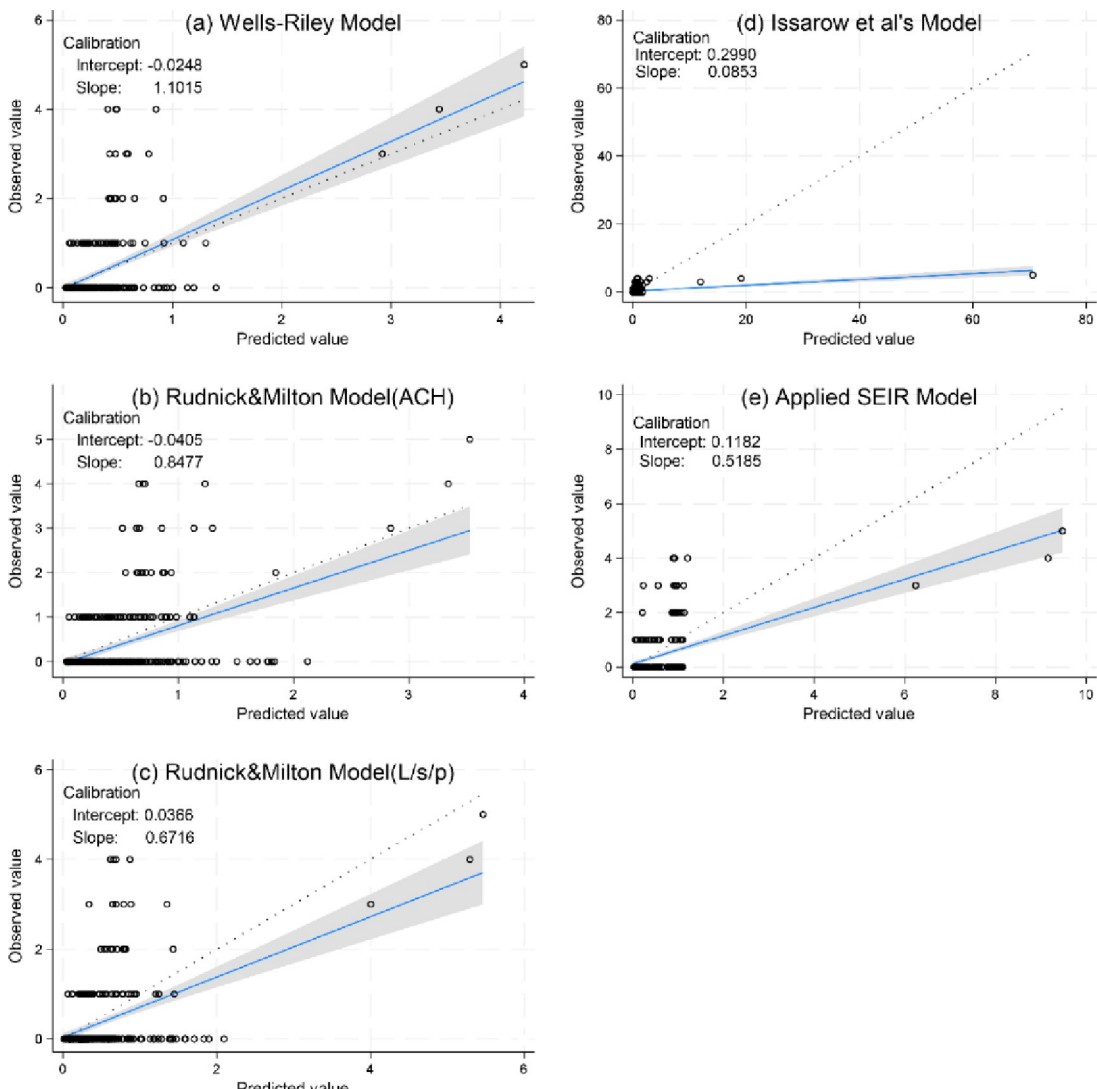

**Fig 4. Calibration plots depicting the external validation of the five dynamic models.**

performance in the internal and external validations despite its ranking as the second-best model in the internal validation (S4 and S5 Tables).

## Discussion

This prospective cohort study employed a standard assessment procedure to validate five dynamic models for TB transmission in predicting PTB incidence in a real prison setting in Thailand. The results revealed that, while the TB transmission probabilities estimated by all five models were significantly and positively associated with the future PTB IR among the inmates, the classic Wells–Riley model exhibited the best fit for predicting the future PTB IR among inmates. In addition to demonstrating the best performance in the in-sample (internal validity), it also exhibited robustness when applied to the out-of-sample group, which had significantly different architectural characteristics and the composition of the cell inmates (external validity). These results were further corroborated using sensitivity analysis, which switched

the dataset between the in-sample and out-of-sample and reanalyzed the data based on the standard assessment procedure.

When compared to the classical Wells–Riley model, more sophisticated models, such as the two models proposed by Rudnick and Milton, the model created by Issarow et al., and the applied SEIR models, did not generate more parsimonious results when it came to predicting the PTB IR. These models possessed more progressive levels of refinement or modification than the classical Wells–Riley model. This was the case even though all the models similarly utilized the number of TB cases in cells and the ventilation rate as the parameters in the model formulation. This outcome does not indicate a conceptual deficiency in the formulation of the four modified models. The most plausible reason is that the more sophisticated models rely on more model parameters, which increases the error variance from imprecise measurements, especially in large-scale surveys. This was particularly evident in the SEIR model, which employed 19 parameters. In contrast, in the classical Wells–Riley model, the two Rudnick & Milton-proposed models, and the Issarow et al. model, the number of parameters was only 7, 9, and 10, respectively. Additionally, the challenge of directly measuring some physiological and PTB clinically related parameters in a field setting necessitated the acquisition of their values from published literature. For example, the numbers of these parameters were 3, 4, 4 and 10 for the classical Wells–Riley model, the two models proposed by Rudnick & Milton, the model developed by Issarow et al., and the applied SEIR model, respectively, and the same values were used for all study units (cells). This could have decreased the power to distinguish between PTB IRs by resulting in less variation in the observed data points for the estimated TB transmission probabilities. With respect to the distinction between the two Rudnick & Milton-proposed models, the investigators, one of whom was a qualified professional industrial hygienist, collected the ventilation rate data in ACH, whereas a small group of prisoners who had only received a few hours of training collected the ventilation rate data in L/s/p. This may have contributed to the latter data's inferior quality and the lower parsimoniousness of the Rudnick & Milton (L/s/p) model when compared to the ACH model. More research is required to ascertain whether these results are applicable for smaller-scale surveys, where field data collection will be more precise and more physiological and PTB clinically related parameters can be obtained directly from the prisoners.

Among the potential challenges to external validity in epidemiological research [31], the ones that were relevant to this study were the differential exposure measurement error, context-induced model miss-formulation, extrapolation beyond the observed data, setting differences, and uncontrolled confounding. First, some model parameters, such as ventilation rate data in L/s/p, cell volume, the number and size of portable fans, and the size of the opening facing prevailing winds, were gathered by various groups of prisoners in different prisons. In addition, the model parameters, including the time-to-TB diagnosis, the number of inmates in the cell, the inmate turnover rate, and the number of overall TB cases in the cell and zone, were derived from the recorded data sources, which were of varying quality across prisons. These could have led to a differential measurement error of the baseline TB transmission probability between the in-sample and the out-of-sample. Second, our post-hoc analysis demonstrated that the ventilation rate in the ACH unit, which was a parameter of all dynamic models except for the Rudnick & Milton (L/s/p) model, was the most effective predictor of future PTB incidence (S6 Table). Including additional parameters with erroneous values in the model formulas is not only unhelpful in finer differentiation of the baseline TB transmission probability, but it may also have a detrimental effect on the predictive power of the ventilation (ACH) parameter with a degree commensurate to the number of the added parameters. In particular, this was the case for parameters with differential measurement errors, as mentioned in the first threat. The physiological and PTB clinically related parameters, which were valued from

published literature rather than directly from each individual inmate, were less affected. We called this threat "context-induced model miss-formulation' on the presumption that it will disappear if the values of the added parameters are more and equally accurate. The dynamic models relying on a higher number of these parameters, such as Issarow et al.'s and applied SEIR models, are particularly susceptible to these two threats. Third, the observed values of baseline TB transmission probabilities of the out-of-sample were significantly higher than those of the in-sample, particularly for the Issarow et al.'s and the two Rudnick & Milton's models (Tables 2 and 4), while Fig 2 shows that the extrapolation of PTB incidence beyond the observed range of baseline TB transmission probabilities of the in-sample encountered significant uncertainty with the magnitude proportionate to the distance further from the upper range of the in-sample. These are, with the exception of the Wells–Riley model, the most likely explanations for the substantially diminished performance of nearly all dynamic models, particularly the Issarow et al. model, when applied to the out-of-sample set.

The unidentified risk or protective factors of PTB incidence that were not captured by the parameters used in the dynamic models may have been present in the marked architectural or structural difference of cells in the out-of-sample from those in the in-sample. This may have resulted in the poorer performance of the equation models fitted in the in-sample when applying them to the out-of-sample. Furthermore, due to practical and ethical reasons, individual-level data such as personal demographics (e.g., gender and age), health history (e.g., personal history of chronic respiratory diseases, type 2 diabetes, immune compromised status, etc.), and health-risk behaviors (e.g., cigarette smoking, alcohol drinking, illegal drug use) could not be collected from each inmate. If the composition of these characteristics varies between the in-sample and the out-of-sample, the external validity may have performed poorly. However, these two threats should have had an equal impact on all dynamic models. In addition to jeopardizing the external validity, all these threats could have also impaired the internal validity of the dynamic models, as indicated by the poor result of the Chi-squared goodness-of-fit test (S1 Table).

Although the presence of these threats suggested a flaw in the design of our validation study, it also presented an opportunity to demonstrate the Wells–Riley model's resilience in the face of these external validity threats. The Wells–Riley model's resilience was maintained even after the dataset was switched between the in-sample and out-of-sample during sensitivity analysis. Because of the lack of a prior study on the performance of these dynamic models in predicting the occurrence of future PTB in the real world, additional research in different prison setting or using different dataset is required to confirm or refute our findings regarding their internal and external validity. In addition, it is necessary to conduct detailed investigations into the potential explanations for these findings, including the practical (as opposed to theoretical) validity of the dynamic model formulas and the appropriateness of the validation research design and conduct.

## Strengths and limitations

The longitudinal design of this research, which lasted for approximately one year, is the first of several advantages. Unlike earlier studies that employed simulations [10, 12, 14, 15, 18], this research explored the relationship between the probability of TB transmission in five dynamic models and future tuberculosis incidence within actual prison cells. Other strengths of the study included a substantial sample size, encompassing many cells across three distinct prisons with varying architectural, administrative, and geographical characteristics. This ensures sufficient statistical power and generalizability of the study findings. Furthermore, standard procedures were employed for assessing the validity of the dynamic models [32]. However, this

study also has certain limitations. First, the "time-to-TB diagnosis" and "inmate turnover rate" metrics were collected via secondary data at the prison zone level, rather than directly from primary data (individual inmates). Second, the trained inmate team measured the absolute ventilation rate in liters per second per person (L/s/p) in the morning following a minimum of 13 h of confinement. This was calculated on the basis of the $CO_2$ concentrations within and outside each cell. However, this method may be imprecise because multiple inmates collected the data. Furthermore, certain parameters related to TB progression were sourced from published literature rather than from local data from the Thai population. These three limitations may have led to inaccuracies in estimating the number of new cases. Lastly, the study period coincided with the Coronavirus disease 2019 pandemic, which may have influenced the number of new TB cases associated with prison and health management.

## Conclusions

As determined by a standard procedure for evaluating the performance of the five dynamic models in predicting future PTB incidence in prison settings, our study demonstrated that the classic Wells–Riley model exhibited the best performance both in the internal and external validations. Given its requirement for fewer parameters, the Wells–Riley model is therefore recommended as the most appropriate dynamic model, especially for large-scale investigations. However, all the assessed dynamic models still exhibited poor fit in their ability to explain the variation of future PTB incidence in the analytical models. Consequently, additional research is required to confirm our findings and to acquire more detailed information that can be used as input to enhance the performance of these dynamic models.

## Supporting information

**S1 Fig. Distribution of the PTB incidence rate based on the TB transmission probability in the observed data range estimated by the five dynamic models.** The green scatter plots indicate observed data for individual prison cells. The dark blue line indicates the fitted line. The two vertical dashed lines represent the observed data interval.
(DOCX)

**S1 Table. Akaike information criterion (AIC) and the Bayesian information criterion (BIC) values of the examined count models in determining the association of baseline TB transmission probability with the PTB incidence.**
(DOCX)

**S2 Table. Comparison of mean predicted projected probabilities and observed proportions of each count of PTB cases in five dynamic models.**
(DOCX)

**S3 Table. Fitted models of the negative binomial regression (NBRG) for the five dynamic models in the sensitivity analysis (n = 333).**
(DOCX)

**S4 Table. In-sample goodness-of-fit and discrimination test results of five dynamic models in the sensitivity analysis (in-sample, n = 333).**
(DOCX)

**S5 Table. Comparison of accuracy and bias metrics from external validation in the sensitivity analysis (out-of-sample, n = 652).**
(DOCX)

**S6 Table. TB incidence rate ratio (IRR) from the NBRM demonstrating effect of each variable in the model on PTB incidence rate predicted by five prediction models ($n$ = 985).** (DOCX)

## Acknowledgments

The investigators would like to thank all stakeholders in the three prisons for their assistance in data collection and the survey. The investigators wish to thank the Department of Corrections at the Ministry of Justice for providing permission for this study. Finally, the authors would like to thank Enago (www.enago.com) for the English language review.

## Author Contributions

**Conceptualization:** Nithinan Mahawan, Wiroj Jiamjarasrangsi.

**Data curation:** Nithinan Mahawan, Wiroj Jiamjarasrangsi.

**Formal analysis:** Nithinan Mahawan, Wiroj Jiamjarasrangsi.

**Funding acquisition:** Nithinan Mahawan, Wiroj Jiamjarasrangsi.

**Investigation:** Nithinan Mahawan, Puchong Sri-Uam, Wiroj Jiamjarasrangsi.

**Methodology:** Nithinan Mahawan, Thanapoom Rattananupong, Puchong Sri-Uam, Wiroj Jiamjarasrangsi.

**Project administration:** Nithinan Mahawan, Wiroj Jiamjarasrangsi.

**Resources:** Nithinan Mahawan, Puchong Sri-Uam, Wiroj Jiamjarasrangsi.

**Software:** Nithinan Mahawan, Wiroj Jiamjarasrangsi.

**Supervision:** Thanapoom Rattananupong, Puchong Sri-Uam, Wiroj Jiamjarasrangsi.

**Validation:** Nithinan Mahawan, Puchong Sri-Uam, Wiroj Jiamjarasrangsi.

**Visualization:** Nithinan Mahawan, Wiroj Jiamjarasrangsi.

**Writing – original draft:** Nithinan Mahawan, Wiroj Jiamjarasrangsi.

**Writing – review & editing:** Thanapoom Rattananupong, Puchong Sri-Uam, Wiroj Jiamjarasrangsi.

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
