## [Decision Letter · Decision Letter 0]

23 Jun 2024

PONE-D-24-03144Validity of five dynamic models of predicting tuberculosis incidence in three prisons in ThailandPLOS ONE

Dear Dr. Jiamjarasrangsi,

Thank you for submitting your manuscript to PLOS ONE. After careful consideration, we feel that it has merit but does not fully meet PLOS ONE’s publication criteria as it currently stands. Therefore, we invite you to submit a revised version of the manuscript that addresses the points raised during the review process.

We look forward to receiving your revised manuscript.

Kind regards,

Martial L Ndeffo-Mbah, Ph.D

Academic Editor

PLOS ONE

Journal Requirements:

"- WJ and NM received the 90th Anniversary of Chulalongkorn University Fund (grant

no. 035, 1/2021) and the FY2021 Thesis Grant for Doctoral Degree Study of the

NationalmResearch Council of Thailand (NRCT; grant no. N41D640002, 2021).

- URI (the 90th Anniversary of Chulalongkorn University Fund):

https://www.grad.chula.ac.th/index.php?lang=th

-URI: https://www.nrct.go.th/"

**Additional Editor Comments:**

Reviewers have raised major concerns about the study design, presentation, and results. Please, thoroughly address these before your manuscript can be given further consideation.

Reviewers' comments:

Reviewer's Responses to Questions

**Comments to the Author**

1. Is the manuscript technically sound, and do the data support the conclusions?

Reviewer #1: Partly

Reviewer #2: Partly

2. Has the statistical analysis been performed appropriately and rigorously? 

Reviewer #1: No

Reviewer #2: Yes

3. Have the authors made all data underlying the findings in their manuscript fully available?

Reviewer #1: No

Reviewer #2: Yes

4. Is the manuscript presented in an intelligible fashion and written in standard English?

Reviewer #1: No

Reviewer #2: Yes

5. Review Comments to the Author

Reviewer #1: Study design was not clearly mentioned in the methodology; the statement of outcome is not clear; Even if the identity of participants are not collected, consent is required you are assessing the PTB status; There is no mention about how the prisoners were tested and who done the medical investigations, etc. The presentation of results in the table could have been better; Rationale is not clear & the objectives are not connected to the rationale. From the stated objective how do you assess the predictive validity of the models under consideration; the Five models under consideration are not mentioned clearly in the introduction or methodology; terminologies were suddenly introduced without the rationale of using the terminologies. e.g. deciles of the probability? why is it occurring unanticipatedly; there were no strong literature for the stated objectives. Lot of Statistical analysis was performed which are not contributing in meeting the objectives.

Reviewer #2: This papers empirically compared 5 dynamic TB transmission predictive models over a one-year prospective cohort study. This is a research more confirming existing knowledge rather than creating new one. It is a good practice with scientific value to challenge and re-confirm scientific theories.

However, I have the following questions / comments for the authors to answer before this paper can be considered further:

- It was said the method was a prospective cohort. However, description of lines 169-177 left me an impression of a case-control study. A condition for cohort is to assure the equivalence of groups at the beginning except the exposure variable as it began with grouping the result into two (with and without new cases) and then investigated into the risk factors. The authors did not detail how the 6 categories were defined and how to justify they were equivalent (in addition to “sufficient number of samples in each category”) and only differ by one variable between each other.

- Line 154-159: how did you manage any conflict between the survey/interview and documents / computer databases?

- Figure 2: I guess the y-axis should be “observed” instead of “predicted”?

- Figure 2: how to explain the hike at decile = 2 for three models?

6. PLOS authors have the option to publish the peer review history of their article (what does this mean?). If published, this will include your full peer review and any attached files.

Reviewer #1: No

Reviewer #2: No

---

## [Author Response · Author response to Decision Letter 0]

23 Aug 2024

Dear Editor and Reviewers,

Thank you very much for your valuable comments. This, however, resulted in the extensive revision of our manuscript for assuring an adequate addressing to your critical comments, with markedly differ from the original manuscript. This may require you to read the whole manuscript again. The responses are as follows.

The Academic Editor:

Comment: Reviewers have raised major concerns about the study design, presentation, and results. Please, thoroughly address these before your manuscript can be given further consideation. 

Response: Thanks for your valuable comment. The study design, presentation, results, and conclusion have been revised in our primary manuscript.

Reviewer 1:

Comment: Study design was not clearly mentioned in the methodology

Response: We have revised the statement about the study design by emphasizing that: (a) each of the 985 cells constitutes a unit of analysis, not individual inmates; (b) baseline TB transmission probability is the exposure or independent variable; and (c) new PTB cases and incidence rate during the follow-up period are the dependent variable or outcome of interest (page 5, lines 172–174).

Comment: The statement of outcome is not clear.

Response: We provide more detail about PTB outcome, as shown on Page 5, Lines 214-236.

Comment: Even if the identity of participants are not collected

Response: We have emphasized that the unit of analysis is the cell, not the individual inmate (page 5, line 174).

We have also addressed this as a limitation of our study, as shown on page 16, lines 541–543.

Comment: Consent is required you are assessing the PTB status

Response: We have provided information that we assessed the PTB status of the inmates indirectly via secondary data, not with the infected inmates (page 6, lines 226–228).

We, however, sought the consent of the superintendent of each prison (page 5, lines 181–187).

Comment: There is no mention about how the prisoners were tested and who done the medical investigations

Response: We have provided the relevant details on page 6, lines 214–223.

Comment: The presentation of results in the table could have been better

Response: Because of the statistical analysis’s elaborate revision, all tables have been modified (Pages 9, 10, 11, 12 and S1 Table)

Comment: Rationale is not clear & the objectives are not connected to the rationale

Response: We have responded to this comment by revising the content in the “Introduction,” as shown on page 2 - 3, lines 61–68, and 71–76.

Comment: From the stated objective how do you assess the predictive validity of the models under consideration

Response: We have employed less promising terms such as “assess the performance of the dynamic models in predicting future PTB incidence,” as shown in the title (page 1, line 2), and the objective (page3, lines 72–76).

We have revised our statistical analysis by using a four-step procedure including model specification tests, in-sample model fitting, internal validation, and external validation (pages 6–8, lines 238–304).

Comment: The Five models under consideration are not mentioned clearly in the introduction or methodology

Response: We have provided a brief description of these models on pages 3–5, lines 79–168.

Comment: Terminologies were suddenly introduced without the rationale of using the terminologies. e.g. deciles of the probability? why is it occurring unanticipatedly

Response: Due to the extensive revision of “the statistical analysis” (pages 6–8, lines 238–304), such terminologies are no longer present in the text.

Comment: There were no strong literature for the stated objectives

Response: We have responded to this comment by revising the content in the “Introduction,” as shown on page 2 - 3, lines 61–68, and 71–76.

Comment: Lot of Statistical analysis was performed which are not contributing in meeting the objectives

Response: We have revised our statistical analysis using a four-step procedure including model specification tests, in-sample model fitting, internal validation, and external validation (pages 6–8, lines 238–304).

We have also extensively revised the results (pages 8–13, lines 306-436), the discussion (pages 14–15, lines 443-447, and 477-530), the conclusion (pages16, lines554-563), and the abstract (page 2, lines 20-40).

Reviewer 2:

Comment: This is a research more confirming existing knowledge rather than creating new one.

Response: To avoid repetition with the existing knowledge, we have deleted the phrase “Objective (b) to determine the association between each parameter of the five dynamic models and the occurrence of PTB in Objective (a)” from our manuscript. 

We focused only on examining "the performance of the dynamic models in predicting future PTB incidence in prison setting" because existing knowledge on this topic is based solely on simulation research; thus, we reexamined it in a real-world situation.

Comment: It is a good practice with scientific value to challenge and re-confirm scientific theories

Response: Our finding challenges the presumption that progressively improved prediction accuracy of the latter models compared to the former models by providing the finding that the original model performed the best in the real-world setting, as shown in the discussion section (pages 14 - 15, lines 443–530), conclusion (pages16, lines554-563), abstract (page 2, lines 20-40)

Comment: It was said the method was a prospective cohort. However, description of lines 169-177 left me an impression of a case-control study.

Response: Due to the extensive revision of “the statistical analysis” (pages 6–8, lines 238–304), such description no longer exists in the manuscript text.

We have revised the table heading from “the comparison of cells with versus without new PRB cases’ to the comparison of the in-sample versus the out-of-sample’ (for the internal and external validations respectively) (pages 9, lines 349-357).

We have also revised the content in the “Results” section on page 8, lines 312–319, to avoid conveying the impression of a case-control study.

Comment: A condition for cohort is to assure the equivalence of groups at the beginning except the exposure variable as it began with grouping the result into two (with and without new cases) and then investigated into the risk factors.

Response: We are unable to perform what you indicated since our exposure of interest (i.e., the baseline TB transmission probability) is a continuous rather than a dichotomous variable. 

We regarded baseline differences among cells in terms of their architectural, demographic, and health history characteristics as unidentified risk factors and potential confounders that we were unable to consider. We have provided a statement about these limitations of our study on page 15, lines 590–521.

Comment: The authors did not detail how the 6 categories were defined and how to justify they were equivalent (in addition to “sufficient number of samples in each category”) and only differ by one variable between each other.

Response: Due to the extensive revision of “the statistical analysis” (pages 6–8, lines 238–304), such categorization of the variables no longer exists in the manuscript text.

Comment: Line 154-159: how did you manage any conflict between the survey/interview and documents / computer databases?

Response: We have revised the statement to make it more clear that we “ascertained PTB cases via the secondary data of the central administrative and medical facilities in the prisons, then sought further information about the cell locations of the PTB inmates (where they live) by the surveys and interviews with zonal staff and inmates who are prison health volunteers” (page 6, lines 226–230).

Comment: Figure 2: I guess the y-axis should be “observed” instead of “predicted”?

Response: Due to the extensive revision of “the statistical analysis” (pages 6–8, lines 238–304) and “Results” sections (pages 8- 13), such mistake in the presentation no longer exists.

Comment: Figure 2: how to explain the hike at decile = 2 for three models?

Response: Due to the extensive revision of “the statistical analysis” (pages 6–8, lines 238–304) and “Results” sections (pages 8- 13), such findings are no longer present in the text.

---

## [Decision Letter · Decision Letter 1]

13 Nov 2024

PONE-D-24-03144R1Performance of five dynamic models in predicting tuberculosis incidence in three prisons in ThailandPLOS ONE

Dear Dr. Jiamjarasrangsi,

Thank you for submitting your manuscript to PLOS ONE. After careful consideration, we feel that it has merit but does not fully meet PLOS ONE’s publication criteria as it currently stands. Therefore, we invite you to submit a revised version of the manuscript that addresses the points raised during the review process.

We look forward to receiving your revised manuscript.

Kind regards,

Martial L Ndeffo-Mbah, Ph.D

Academic Editor

PLOS ONE

Additional Editor Comments:

Thank you very much for addressing the reviewers comments. However, new concerns have been raised by the reviewers which need to be addressed before the manuscript can be given further consideration.

Reviewers' comments:

Reviewer's Responses to Questions

**Comments to the Author**

1. If the authors have adequately addressed your comments raised in a previous round of review and you feel that this manuscript is now acceptable for publication, you may indicate that here to bypass the “Comments to the Author” section, enter your conflict of interest statement in the “Confidential to Editor” section, and submit your "Accept" recommendation.

Reviewer #3: All comments have been addressed

Reviewer #4: (No Response)

2. Is the manuscript technically sound, and do the data support the conclusions?

Reviewer #3: Yes

Reviewer #4: Partly

3. Has the statistical analysis been performed appropriately and rigorously? 

Reviewer #3: Yes

Reviewer #4: I Don't Know

4. Have the authors made all data underlying the findings in their manuscript fully available?

Reviewer #3: Yes

Reviewer #4: No

5. Is the manuscript presented in an intelligible fashion and written in standard English?

Reviewer #3: Yes

Reviewer #4: Yes

6. Review Comments to the Author

Reviewer #3: The revision of study design and results have addressed previous comments. This study provides a comparison of five dynamic models for PTB incidence in a prison setting. The authors offer good insights by evaluating the performance of the Wells–Riley equation, two models proposed by Rudnick & Milton, the Issarow et al. model, and the SEIR TB transmission model.

Reviewer #4: The authors have made substantial revisions from previous comments (of which I was not involved). I have reviewed this manuscript as a whole rather than just checking if the previous revisions were addressed.

The authors aim to compare the predictive performance of five different models, building on their previous work published earlier this year. They found the best performing model, but ultimately concluded that further research is needed to support their results.

Some minor comments:

- The readability of the paper can be improved - it felt like the paper had not been proof-read before submission.

- The presentation of the models I believe should be in the methods section rather than the introduction section.

- Variables explained in the text in relation to the models' equations are not typed in 'math' mode and thus make it hard to follow. Furthermore, some variables are repeated (given most of the models are extensions of others surely there is a better way to explain the parameters without repetition).

- Line 114 - parameter \\theta is mentioned but it is not actually present in the equation above.

- Line 117 - 'per person' mentioned twice.

- There is generally an inconsistency in how units are presented (e.g. per hour vs hr^{-1})

- Line 132 - first define \\beta and \\mu and then define (\\beta - \\mu).

- Line 139 - parameter p defined twice?

- I am a bit confused about the SEIR model and the units used (e.g person/year and person/180 days etc. shouldn't there be consistency?)

- Line 177 - change "5" to "five" to be consistent with the rest of the text.

- Line 233 - IR already defined in introduction so can use abbreviation. Also in the text incidence rate is sometimes mentioned via the abbreviation and sometimes written out in full. Why introduce the abbreviation if you will not use it consistently?

- The prisons are introduced as "Prison A" etc. and then referred to as "prison A" etc. Be consistent with capitalisation.

Major comments:

- In the statistical analysis section, why was prison B the in-sample and the prisons A & C out-of-sample? There is no explanation given for this choice. Have you tried other combinations for robustness?

- Did you consider other regression models e.g. zero-inflation or hurdle models with the negative binomial to consider for excess zeros? The choice of model is not really justified much.

- As some of the parameters used were obtained from secondary data and from the literature, I believe sensitivity analysis would be useful to assess robustness or identify critical parameters.

Main concern:

- I believe the main strength of this paper is that it uses real-data (for the most part) rather than simulations that studies in the past have done. However, the novelty introduced here is minimal. The authors themselves say that their results need further research to confirm or refute them. I think this paper would be much stronger if this 'further research' mentioned was done to support the results or at least describe in detail what 'further research' could be.

7. PLOS authors have the option to publish the peer review history of their article (what does this mean?). If published, this will include your full peer review and any attached files.

Reviewer #3: No

Reviewer #4: No

---

## [Author Response · Author response to Decision Letter 1]

11 Dec 2024

Dear Editor and Reviewers,

Thank you for your constructive comments that both stimulated and guided us in improving our manuscript, which had also enhanced our research knowledge and skill particularly in statistical analytical aspect. The responses are as follows.

The Editor:

Comment: Thank you very much for addressing the reviewers comments. However, new concerns have been raised by the reviewers which need to be addressed before the manuscript can be given further consideration.

Response: Thanks for your valuable comment. All new concerns have been revised in our primary manuscript.

Reviewer #3:

Thank you very much sir/madam for your favorable opinion.

Reviewer #4:

Minor Comments

Comment: The readability of the paper can be improved - it felt like the paper had not been proof-read before submission.

Response: Your comments are greatly appreciated. We had meticulously proofread the manuscript before submission.

Comment: The presentation of the models I believe should be in the methods section rather than the introduction section.

Response: We followed your recommendation, as illustrated on lines 79–167 of pages 4–7.

Comment: Variables explained in the text in relation to the models' equations are not typed in 'math' mode and thus make it hard to follow. 

Response: Thank you for your feedback. In accordance with your suggestion, the models’ equations have been modified to be in the “math” mode, as demonstrated on page 4, lines 89, page 5, line 110 and on page 6, line 130.

Comment: Furthermore, some variables are repeated (given most of the models are extensions of others surely there is a better way to explain the parameters without repetition).

Response: Although certain variables are repeated or represented using the same symbols, there are differences in the details and methods employed to calculate the probability of tuberculosis transmission in each model. Our approach is consistent with the definitions outlined in the original research articles. For instance, the variable ventilation rate in Rudnick & Milton’s modified model can be denoted in two distinct ways: air changes per hour (ACH) or liters per second per person. Conversely, other models in this study exclusively use ACH. Similarly, the unit for the variable t in the classic Wells–Riley model is expressed in months, whereas the unit for the variable t is in days in Rudnick & Milton’s modified model. Furthermore, the variable t is not included in the SEIR model.

Therefore, we continue to present the detailed specifications of each model to ensure that readers or researchers intending to utilize the information from this study can do so without any difficulty or confusion. The dataset files, which are accessible from the Figshare repository (DOI: 10.6084/m9.figshare.26787874), contain the calculation methodologies. However, we have removed specific variables that were previously described in the Wells–Riley model.

Comment: Line 114 - parameter \\theta is mentioned but it is not actually present in the equation above.

Response: In accordance with your comment, we have made the necessary revisions to the content on page 5, line 110.

Comment: Line 117 - 'per person' mentioned twice.

Response: As per your recommendation, we have made the necessary revisions to line 117 on page 6.

Comment: There is generally an inconsistency in how units are presented (e.g. per hour vs hr^{-1})

Response: We have replaced the terms “hr” with “h” and “hr−1” with “/hour,” as shown on Page 6, lines 135, 137 and 138 respectively.

Comment: Line 132 - first define \\beta and \\mu and then define (\\beta - \\mu).

Response: We implemented your recommendation, as illustrated on lines 132–135 of page 6.

Comment: Line 139 - parameter p defined twice?

Response: The parameter p is defined twice, as the second instance of p is a sub-variable used in the calculation of the variable ƒ (the fraction of rebreathed air).

Comment: I am a bit confused about the SEIR model and the units used (e.g person/year and person/180 days etc. shouldn't there be consistency?)

Response: The variable number of overall TB cases (infectious) based on actual data has not been utilized in any previous studies, as indicated by the review of related literature. Instead, most studies have relied on assumed values to determine the probability of TB transmission. In contrast, this study monitored TB prevalence and incidence in a real-world setting over a 6-month period (equivalent to 180 days), which is consistent with the standard duration of TB treatment in Thailand prisons. This 180-day or 6-month period was also employed in our calculations of the probability of TB transmission, in accordance with the original studies.

Comment: Line 177 - change "5" to "five" to be consistent with the rest of the text.

Response: We have changed "5" to "five” on page 8, line 175, as per your suggestion. 

Comment: Line 233 - IR already defined in introduction so can use abbreviation. Also in the text incidence rate is sometimes mentioned via the abbreviation and sometimes written out in full. Why introduce the abbreviation if you will not use it consistently?

Response: We have revised the content in line with your comment, as shown on page 10, line 230; page 12, line 271; page16, lines 346 and 347; page 17, line 367, 375 and 380; page 23, line 429, 430 and 440; page 23, line 455.

Comment: The prisons are introduced as "Prison A" etc. and then referred to as "prison A" etc. Be consistent with capitalisation.

Response: We have made the necessary modification, as per your recommendation.

Major Comments

Comment: Rationale for selecting the in-sample and out-of-sample

Response: The rationale for selecting the in-sample and out-of-sample is outlined in the “Statistical analysis” section on page 10 and 11, lines 236–243.

Comment: Other combination of the in-sample and out-of-sample

Response: We conducted the sensitivity analysis by reanalyzing the data after switching the datasets between the in-sample and out-of-sample. We present the statements/contents regarding this activity in:

 - The “Statistical analysis” section on page 13, lines 306–310;

 - The “sensitivity analysis” sub-section of the “Results” section on page 22, lines 417–424, and in Tables S3 to S6;

 - The “Discussion” section on page 23, lines 434–436.

Comment: Rationale for selecting count models

Response: The rationale and procedure for selecting the optimal count model are detailed in the “Model specification test” sub-section” of the “Statistical analysis” section on page 11, lines 254–262, and in Table S1.

To execute model fitting, we employed ZIP, ZINB, and NBRG. The results are quite comparable, as illustrated below. Therefore, we incorporated only the NBRG results in the manuscript.

 Table Fitted models for the five dynamic models (n = 652)

Prediction Model Constant (b0) Beta (b1) SD IRR (95%CI)

 NBRG 

Wells–Riley −5.067 0.030 0.006 1.030 (1.018, 1.043)

Rudnick & Milton(ACH) −5.116 0.051 0.013 1.053 (1.025, 1.081)

Rudnick & Milton(L/s/p) −4.823 0.067 0.028 1.069 (1.011, 1.130)

Issarow et al. −4.875 0.033 0.009 1.033 (1.016, 1.051)

Applied SEIR −5.544 0.031 0.010 1.031 (1.011, 1.051)

 ZIP 

Wells–Riley −4.693 0.028 0.006 1.028 (1.017, 1.040)

Rudnick & Milton(ACH) −4.647 0.050 0.013 1.051 (1.024, 1.078)

Rudnick & Milton(L/s/p) −4.271 0.056 0.026 1.057 (1.004, 1.114)

Issarow et al. −4.450 0.029 0.007 1.030 (1.015, 1.045)

Applied SEIR −5.064 0.031 0.010 1.031 (1.011, 1.052)

 ZINB 

Wells–Riley −4.692 0.028 0.006 1.028 (1.016, 1.040)

Rudnick & Milton(ACH) −4.647 0.050 0.013 1.051 (1.024, 1.078)

Rudnick & Milton(L/s/p) −4.271 0.056 0.027 1.058 (1.004, 1.114)

Issarow et al. −4.450 0.029 0.007 1.030 (1.015, 1.045)

Applied SEIR −5.064 0.031 0.010 1.031 (1.011, 1.052)

CI = Confidence interval, IRR = Incidence rate ratio, SD = standard deviation

Comment: Sensitivity analysis to assess robustness or identify critical parameters.

Response: We conducted post-hoc analysis for substantiating our statement that the ventilation rate in ACH is a critical parameter in predicting future PTB incidence (“Discussion” section, page 25, lines 476–479), and the results are presented in Table S6.

Main Concerns

Comment: Comment about ‘further research’

Response: The “further research” comprised the previously described investigations during sensitivity analysis, which involved switching the datasets between the in-sample and out-of-sample.

Based on the result from the sensitivity analysis, we also incorporated the statement to further support the conclusion regarding the Wells–Riley model’s resilience, as demonstrated in the “Discussion” section on page 26, lines 512–516.

We have however retained the statement regarding “further research” and provided a more detailed explanation, as illustrated in the “Discussion” section on page 26, lines 516–522.

---

## [Decision Letter · Decision Letter 2]

10 Jan 2025

Performance of five dynamic models in predicting tuberculosis incidence in three prisons in Thailand

PONE-D-24-03144R2

Dear Dr. Jiamjarasrangsi,

We’re pleased to inform you that your manuscript has been judged scientifically suitable for publication and will be formally accepted for publication once it meets all outstanding technical requirements.

Kind regards,

Martial L Ndeffo-Mbah, Ph.D

Academic Editor

PLOS ONE

Additional Editor Comments (optional):

Reviewers' comments:

Reviewer's Responses to Questions

**Comments to the Author**

1. If the authors have adequately addressed your comments raised in a previous round of review and you feel that this manuscript is now acceptable for publication, you may indicate that here to bypass the “Comments to the Author” section, enter your conflict of interest statement in the “Confidential to Editor” section, and submit your "Accept" recommendation.

Reviewer #4: All comments have been addressed

2. Is the manuscript technically sound, and do the data support the conclusions?

Reviewer #4: Yes

3. Has the statistical analysis been performed appropriately and rigorously? 

Reviewer #4: Yes

4. Have the authors made all data underlying the findings in their manuscript fully available?

Reviewer #4: Yes

5. Is the manuscript presented in an intelligible fashion and written in standard English?

Reviewer #4: Yes

6. Review Comments to the Author

Reviewer #4: The revisions I suggested in the previous round have been addressed and I am happy to change my recommendation to accept for publication if the editors also agree.

7. PLOS authors have the option to publish the peer review history of their article (what does this mean?). If published, this will include your full peer review and any attached files.

Reviewer #4: No

---

## [Editor Report · Acceptance letter]

14 Jan 2025

PONE-D-24-03144R2 

PLOS ONE

Dear Dr. Jiamjarasrangsi, 

I'm pleased to inform you that your manuscript has been deemed suitable for publication in PLOS ONE. Congratulations! Your manuscript is now being handed over to our production team.

Kind regards, 

on behalf of

Dr. Martial L Ndeffo-Mbah 

Academic Editor

PLOS ONE